# DITTO: OFFLINE IMITATION LEARNING WITH WORLD MODELS

## ABSTRACT

We propose DITTO, an offline imitation learning algorithm which uses world models and on-policy reinforcement learning to addresses the problem of covariate shift, without access to an oracle or any additional online interactions. We discuss how world models enable offline, on-policy imitation learning, and propose a simple intrinsic reward defined in the world model latent space that induces imitation learning by reinforcement learning. Theoretically, we show that our formulation induces a divergence bound between expert and learner, in turn bounding the difference in reward. We test our method on difficult Atari environments from pixels alone, and achieve state-of-the-art performance in the offline setting.

## 1 INTRODUCTION

Generating agents which can capably act in complex environments is challenging. In the most difficult environments, hand-designed controllers are often insufficient and learning-based methods must be used to achieve good performance. When we can exactly specify the goals and constraints of the problem using a reward function, reinforcement learning (RL) offers an approach which has been extremely successful at solving a range of complex tasks, such as the strategy games of Go (Silver et al., 2016), Starcraft (Vinyals et al., 2019), and poker (Brown et al., 2020), and difficult real world control problems like quadrupedal locomotion (Lee et al., 2020), datacenter cooling (Lazic et al., 2018), and chip placement (Mirhoseini et al., 2020). RL lifts the human designer's work from explicitly designing a good policy, to designing a good reward function. However, this optimization often results in policies that maximize reward in undesirable ways that their designers did not intend (Lehman et al., 2018) - the so-called *reward hacking* phenomenon (Krakovna et al., 2020). To combat this, practitioners spend substantial effort observing agent failure modes and tuning reward functions with extra regularization terms and weighting hyperparameters to counteract undesirable behaviors (Wang et al., 2022) (Peng et al., 2017).

Imitation learning (IL) offers an alternative approach to policy learning which bypasses reward specification by directly mimicking the behavior of an expert demonstrator. The simplest kind of IL, behavior cloning (BC), trains an agent to predict an expert's actions from observations, then acts on these predictions at test time. This approach fails to account for the sequential nature of decision problems, since decisions at the current step affect which states are seen later. The distribution of states seen at test time will differ from those seen during training unless the expert training data covers the entire state space, and the agent makes no mistakes. This distribution mismatch, or covariate shift, leads to a compounding error problem: initially small prediction errors lead to small changes in state distribution, which lead to larger errors, and eventual departure from the training distribution altogether (Pomerleau, 1989). Intuitively, the agent has not learned how to act under its own induced distribution. This was formalized in the seminal work of Ross & Bagnell (2010), who gave a tight regret bound on the difference in return achieved by expert and learner, which is quadratic in the episode length for BC.

Follow-up work in Ross et al. (2011) showed that a linear bound on regret can be achieved if the agent learns online in an interactive setting with the expert: Since the agent is trained *under its own distribution* with expert corrections, there is no distribution mismatch at test-time. This works well when online learning is safe and expert supervision can be scaled, but is untenable in many real-world use-cases such as robotics, where online can be unsafe, time-consuming, or otherwise

infeasible. On the one hand, we want to generate data on-policy to avoid covariate shift, but on the other hand, we may not be able to afford to learn online due to safety or other concerns.

Ha & Schmidhuber (2018) propose a two-stage approach to policy learning, where agents first learn to predict the environment dynamics with a recurrent neural network called a "world model" (WM), and then learn the policy inside the WM alone. This approach is desirable since it enables on-policy learning *offline*, given the existence of the world model. Similar model-based learning methods have recently achieved success in standard online RL settings (Hafner et al., 2021), and impressive zero-shot transfer of policies trained solely in the WM to physical robots (Wu et al., 2022).

In this paper, we propose an imitation learning algorithm called **Dream Imitation (DITTO)**, which addresses the tension between offline and on-policy learning, by training an agent using on-policy RL inside a learned world model. Specifically, we define a reward which measures divergence between the agent and expert demonstrations in the latent space of the world model, and show that optimizing this reward with RL induces imitation on the expert. We discuss the relationship between our method and the imitation learning as divergence minimization framework (Ghasemipour et al., 2019), and show that our method optimizes a similar bound without requiring adversarial training. We compare our method against behavior cloning and generative adversarial imitation learning (GAIL, Ho & Ermon (2016)), which we adapt to the world model setting, and show that we achieve better performance and sample efficiency in challenging Atari environments from pixels alone.

Our main contributions are summarized as follows:

- We discuss how world models relieve the tension between offline and on-policy learning methods, which mitigates covariate shift from offline learning.

- We demonstrate the first fully offline model-based imitation learning method that achieves strong performance on Atari from pixels, and show that our agent outperforms competitive baselines adapted to the offline setting.

- We show how imitation learning can naturally be cast as a reinforcement learning problem in the latent space of learned world models, and propose a latent-matching intrinsic reward which compares favorably against commonly used adversarial and sparse formulations.

## 2 RELATED WORK

### 2.1 IMITATION BY REINFORCEMENT LEARNING

Imitation learning algorithms can be classified according to the set of resources needed to produce a good policy. Ross et al. (2011) give strong theoretical and empirical results in the online interactive setting, which assumes that we can both learn while acting online in the real environment, and that we can interactively query an expert policy to e.g. provide the learner with the optimal action in the current state. Follow-up works have progressively relaxed the resource assumptions needed to produce good policies. Sasaki & Yamashina (2021) show that the optimal policy can be recovered with a modified form of BC when learning from imperfect demonstrations, given a constraint on the expert sub-optimality bound. Brantley et al. (2020) study covariate shift in the online, non-interactive setting, and demonstrate an approximately linear regret bound by jointly optimizing the BC objective with a novel policy ensemble uncertainty cost, which encourages the learner to return to and stay in the distribution of expert support. They achieve this by augmenting the BC objective with the following uncertainty cost term:

$$\text{Var}_{\pi \sim \Pi_E} (\pi(a|s)) = \frac{1}{E} \sum_{i=1}^{E} (\pi_i(a|s) - \frac{1}{E} \sum_{j=1}^{E} \pi_j(a|s))^2 \tag{1}$$

This term measures the total variance of a policy ensemble $\Pi_E = \{\pi_1, ..., \pi_E\}$ trained on disjoint subsets of the expert data. They optimize the combined BC plus uncertainty objective using standard online RL algorithms, and show that this mitigates covariate shift.

Inverse reinforcement learning (IRL) can achieve improved performance over BC by first learning a reward from the expert demonstrations for which the expert is optimal, then optimizing that reward with on-policy reinforcement learning. This two-step process, which includes on-policy RL in the

second step, helps IRL methods mitigate covariate shift due to train and test distribution mismatches. However, the learned reward function can fail to generalize outside of the distribution of expert states which form its support.

A recent line of work treats IRL as *divergence minimization*: instead of directly copying the expert actions, they minimize a divergence measure between expert and learner state distributions

$$\min_\pi \mathbb{D}\left(\rho^\pi, \rho^E\right) \tag{2}$$

where $\rho^\pi(s, a) = (1 - \gamma) \sum_{t=0}^{\infty} \gamma^t P(s_t = s, a_t = a)$ is the discounted state-action distribution induced by $\pi$, and $\mathbb{D}$ is a divergence measure between probability distributions. The popular GAIL algorithm (Ho & Ermon, 2016) constructs a minimax game in the style of GANs (Goodfellow et al., 2014) between the learner policy $\pi$, and a discriminator $D_\psi$ which learns to distinguish between expert and learner state distributions

$$\max_\pi \min_{D_\psi} \mathbb{E}_{(s,a)\sim\rho^E}\left[-\log D_\psi(s, a)\right] + \mathbb{E}_{(s,a)\sim\rho^\pi}\left[-\log\left(1 - D_\psi(s, a)\right)\right] \tag{3}$$

This formulation minimizes the Jensen-Shannon divergence between the expert and learner policies, and bounds the expected return difference between agent and expert. However, Wang et al. (2019) point out that adversarial reward learning is inherently unstable since the discriminator is always trained to penalize the learner state-action distribution, even if the learner has converged to the expert policy. This finding is consistent with earlier work (Brock et al., 2019) which observed discriminator overfitting, necessitating early stopping to prevent training collapse. Multiple works have reported difficulty getting GAIL to work reliably in pixel-based observation environments (Brantley et al., 2020) (Reddy et al., 2020).

To combat problems with adversarial training, Wang et al. (2019) and Reddy et al. (2020) consider reducing IL to RL on an intrinsic reward

$$r(s, a) = \begin{cases} 1 & \text{if } (s, a) \in \mathcal{D}^E \\ 0 & \text{otherwise} \end{cases} \tag{4}$$

where $\mathcal{D}^E$ is the expert dataset. While this sparse formulation is impractical e.g. in continuous action settings, they show that a generalization of the intrinsic reward using support estimation by random network distillation (Burda et al., 2019) results in stable learning that matches the performance of GAIL without the need for adversarial training. Ciosek (2022) showed that this formulation is equivalent to divergence minimization under the total variation distance, and produced a bound on the difference in extrinsic reward achieved between the expert and a learner trained with this approach.

## 2.2 OFFLINE LEARNING

Kumar et al. (2019) showed how naïve application of Bellman backups in off-policy RL results in incorrect, overly optimistic value estimation. These off-policy bootstrapping errors are further compounded in the model-based setting, since states where either the learned dynamics or reward functions generalize poorly will be found and incorrectly backed out by a naïve Q-learner. This is acceptable in the online setting, since areas of state space which the agent is overly-optimistic about will tend to be visited, resulting in natural corrections to inaccuracies; but the offline setting does not share this property. To counteract these problems, offline RL methods either constrain the learner policy to stay on the expert distribution directly (Wu et al., 2019), or use pessimistic methods to discourage value estimates from becoming too large out of distribution (Kumar et al., 2020) (Rigter et al., 2022).

## 2.3 WORLD MODELS

World models have recently emerged as a promising approach to model-based learning. Ha & Schmidhuber (2018) defined the prototypical two-part model: a variational autoencoder (VAE) is trained to reconstruct observations from individual frames, while a recurrent state-space model (RSSM) is trained to predict the VAE encoding of the next observation, given the current latent state and action. World models can be used to train agents entirely inside the learned latent space,

without the need for expensive decoding back to the observation space. Hafner et al. (2020) introduced Dreamer, an RL agent which is trained purely in the latent space of the WM, and successfully transfers to the true environment at test-time. Wu et al. (2022) showed that the same approach can be used to simultaneously learn a model and agent policy to control a physical quadrupedal robot online, without the control errors usually associated with transferring policies trained only in simulation to a physical system (Hwangbo et al., 2019).

In this work, we propose the use of world models to address a number of common problems in imitation learning. Intrinsic rewards which induce imitation learning, like those introduced in Reddy et al. (2020) and Wang et al. (2019), can pose challenging online learning problems, since the rewards are sparse or require tricky additional training procedures to work in high-dimensional observation spaces. Similarly, approaches like GAIL (Ho & Ermon, 2016) and AIRL (Fu et al., 2018) require adversarial on-policy training that is difficult to make work in practice. In contrast, our approach remedies both the online learning and reward specification problems by performing safe offline policy learning solely inside the compact latent space of the world model, and uses a natural divergence measure as reward: distance between learner and expert in the world model latent space. This provides a conceptually simple and dense reward signal for imitation by reinforcement learning, which we find outperforms competitive approaches in data efficiency and asymptotic performance.

## 3 DREAM IMITATION

We study imitation learning in a partially observable Markov decision process (POMDP) with discrete time-steps and actions, and high dimensional observations generated by an unknown environment. The POMDP $\mathcal{M}$ is composed of the tuple $\mathcal{M} = (\mathcal{S}, \mathcal{A}, \mathcal{X}, \mathcal{R}, \mathcal{T}, \mathcal{U}, \gamma)$, where $s \in \mathcal{S}$ is the state space, $a \in \mathcal{A}$ is the action space, $x \in \mathcal{X}$ is the observation space, $\gamma$ is the discount factor, and $r = \mathcal{R}(s, a)$ is the reward function. The transition dynamics are Markovian, and given by $s_{t+1} \sim \mathcal{T}(\cdot \mid s_t, a_t)$. The agent does not have access to the underlying states, and only receives observations represented by $x_t \sim \mathcal{U}(\cdot \mid s)$. The goal is to maximize the discounted sum of extrinsic (environment) rewards $\mathbb{E}[\Sigma_t \gamma^t r_t]$, which the agent does not have access to.

Training proceeds in two parts: we first learn a world model from recorded sequences of observations, then train an actor-critic agent to imitate the expert in the world model. The latent dynamics of the world model define a fully observable Markov decision process (MDP), since the model states $\hat{s}_t$ are Markovian. Model-based rollouts always begin from an observation drawn from the expert demonstrations, and continue for a fixed set of time steps $H$, the agent training horizon. The agent is rewarded for matching the latent trajectory of the expert.

### 3.1 PRELIMINARIES

We show that bounding the learner-expert state distribution divergence in the world model also bounds their return difference in the actual environment, and connect our method to the *IL as divergence minimization* framework (Ghasemipour et al., 2019). Rafailov et al. (2021) showed that for a learned dynamics model $\widehat{\mathcal{T}}$ whose total variation from the true transitions is bounded such that $\mathbb{D}_{\text{TV}}(\mathcal{T}(s, a), \widehat{\mathcal{T}}(s, a)) \leq \alpha \quad \forall (s, a) \in \mathcal{S} \times \mathcal{A}$ and $R_{\max} = \max_{(s,a)} \mathcal{R}(s, a)$ then

$$\left| \mathcal{J}(\pi^E, \mathcal{M}) - \mathcal{J}(\pi, \mathcal{M}) \right| \leq \underbrace{\alpha \frac{R_{\max}}{(1 - \gamma)^2}}_{\text{learning error}} + \underbrace{\frac{R_{\max}}{1 - \gamma} \mathbb{D}_{\text{TV}}\left( \rho_{\mathcal{M}}^E, \rho_{\widehat{\mathcal{M}}}^\pi \right)}_{\text{adaptation error}} \tag{5}$$

where $\mathcal{J}(\pi, \mathcal{M})$ is the expected return of policy $\pi$ in MDP $\mathcal{M}$, and $\widehat{\mathcal{M}}$ is the "imagination MDP" induced by the world model. This implies the difference between the expert return and the learner return in the true environment is bounded by two terms, 1) a term proportional to the model approximation error $\alpha$, which could in principle be reduced with more data, and 2) a model domain adaptation error term, which captures the generalization error of a model trained under data from one policy, and deployed under another. Rafailov et al. (2021) also show that bounding the divergence between *latent* distributions upper bounds the true state distribution divergence. Formally, given a latent representation of the transition history $z_t = q(x_{\leq t}, a_{<t})$ and a belief distribution $P(s_t \mid x_{\leq t}, a_{<t}) = P(s_t \mid z_t)$, then if the policy conditions only on the latent representation $z_t$ such that the belief distribution is independent of the current action $P(s_t \mid z_t, a_t) = P(s_t \mid z_t)$,

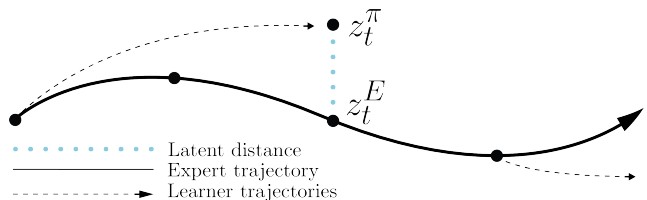

Figure 1: The learner begins from random expert latent states during training, and generates on-policy latent trajectories in the world model. The intrinsic reward 8 encourages the learner to recover from its mistakes over multiple time steps to match the expert trajectory.

then the divergence between the latent state distribution of the expert and learner upper bounds the divergence between their true state distribution:

$$\mathbb{D}_f(\rho_{\mathcal{M}}^{\pi}(x,a) \parallel \rho_{\mathcal{M}}^{E}(x,a)) \leq \mathbb{D}_f(\rho_{\mathcal{M}}^{\pi}(s,a) \parallel \rho_{\mathcal{M}}^{E}(s,a)) \leq \mathbb{D}_f(\rho_{\mathcal{M}}^{\pi}(z,a) \parallel \rho_{\mathcal{M}}^{E}(z,a)) \quad (6)$$

Where $\mathbb{D}_f$ is a generic $f$-divergence , e.g. KL or TV. This result, along with equation 5, suggests that minimizing divergence in the model latent space is sufficient to bound the expected expert-learner return difference.

**Reward**     To bound expert-learner state distribution divergences, prior approaches have focused on sparse indicator function rewards (Ciosek, 2022), or adversarial reward learning (Ghasemipour et al., 2019). We propose a new formulation, which rewards the agent for matching the expert latent state-action pairs over an episode. In particular, for an arbitrary distance function $d$, agent state-action latent $z_t^{\pi}$, and a set of expert state-action latents $\mathcal{D}_E$:

$$r_t^{int}(z_t^{\pi}) = 1 - \min_{z^E \in \mathcal{D}_E} d(z_t^{\pi}, z^E) \quad (7)$$

This function rewards matching the agent's state-action pairs to the expert's, as studied in Ciosek (2022). The major difference is that we also smooth in the latent space, meaning an exact match isn't required for a reward. We show in appendix A how to make this relaxed reward compatible with the theoretical results from Ciosek (2022). In particular, we prove that maximizing this reward bounds the total variation in latent-state distributions between the expert and learner, as well as bounding their extrinsic reward difference.

Intuitively, matching latent states between the learner and expert is easier than matching observations, since the representations learned from generative world model training should provide a much richer signal of state similarity. In practice, the minimization over $\mathcal{D}_E$ can be computationally expensive, so we modify the objective 7 to exactly match learner latent states to expert latents from the same time-step, as shown in Figure 1. In particular, we randomly sample consecutive expert latents $z_{t:t+H}^E$ from $\mathcal{D}_E$ and unroll the agent from the same starting state in the world model, yielding a sequence of agent latents $z_{t:t+H}^{\pi}$. Finally, we compute a reward at each step $t$ as follows:

$$r_t^{int}(z_t^E, z_t^{\pi}) = 1 - d(z_t^E, z_t^{\pi}) = \frac{z_t^E \cdot z_t^{\pi}}{\max(\|z_t^E\|, \|z_t^{\pi}\|)^2} \quad (8)$$

This formulation changes our method from distribution matching to mode seeking, since states frequently visited by the expert will receive greater reward in expectation. We found that this modified dot product reward empirically outperformed $L_2$ and cosine-similarity metrics.

### 3.2 WORLD MODEL

**Dataset**     World model training can be performed using datasets generated by policies of any quality, since the model only predicts transition dynamics. The transition dataset is composed of $N$ episodes $e_n$ of sequences of observations $x_t$, actions $a_t$: $\mathcal{D} = \{(x_t, a_t)_{t=0}^{\|e_n\|} \mid n \in N\}$.

**Model architecture**     We adapt the architecture proposed by Hafner et al. (2021), which is composed of an image encoder, a recurrent state-space model (RSSM) which learns the transition dynamics, and a decoder which reconstructs observations from the compact latent states. The encoder

uses a convolutional neural network (CNN) to produce representations, while the decoder is a transposed CNN. The RSSM predicts a sequence of length $T$ deterministic recurrent states $(h_t)_{t=0}^T$, each of which are used to parameterize two distributions over stochastic hidden states. The stochastic posterior state $z_t$ is a function of the current observation $x_t$ and recurrent state $h_t$, while the stochastic prior state $\hat{z}_t$ is trained to match the posterior without access to the current observation. The current observation is reconstructed from the full model state, which is the concatenation of the deterministic and stochastic states $\hat{s}_t = (h_t, z_t)$. The RSSM components are:

$$
\begin{aligned}
&\text{Model state:} && \hat{s}_t = (h_t, z_t) \\
&\text{Recurrent state:} && h_t = f_\phi(\hat{s}_{t-1}, a_{t-1}) \\
&\text{Prior predictor:} && \hat{z}_t \sim p_\phi(\hat{z}_t \mid h_t) \\
&\text{Posterior predictor:} && z_t \sim q_\phi(z_t \mid h_t, x_t) \\
&\text{Image reconstruction:} && \hat{x}_t \sim p_\phi(\hat{x}_t \mid \hat{s}_t)
\end{aligned}
\tag{9}
$$

All components are implemented as neural networks, with a combined parameter vector $\phi$. Since the prior model predicts the current model state using only the previous action and recurrent state, without using the current observation, we can use it to learn behaviors without access to observations or decoding back into observation space. The prior and posterior models predict categorical distributions which are optimized with straight-through gradient estimation (Bengio et al., 2013). All components of the model are trained jointly with a modified ELBO objective:

$$
\min_\phi \mathbb{E}_{q_\phi(z_{1:T} \mid a_{1:T}, x_{1:T})} \left[ \sum_{t=1}^T -\log p_\phi(x_t \mid \hat{s}_t) + \beta D_{\text{KL-B}}(q_\phi(z_t \mid \hat{s}_t) \parallel p_\phi(\hat{z}_t \mid h_t)) \right]
\tag{10}
$$

where $D_{\text{KL-B}}(q \parallel p)$ denotes KL balancing (Hafner et al., 2021), which is used to control the regularization of prior and posterior towards each other with a parameter $\delta$,

$$
D_{\text{KL-B}}(q \parallel p) = \delta \underbrace{D_{\text{KL}}(q \parallel sg(p))}_{\text{posterior regularizer}} + (1 - \delta) \underbrace{D_{\text{KL}}(sg(q) \parallel p)}_{\text{prior regularizer}}
\tag{11}
$$

and $sg(\cdot)$ is the stop gradient operator. The idea behind KL balancing is that the prior and posterior should not be regularized at the same rate: the prior should update more quickly towards the posterior, which encodes strictly more information.

## 3.3 AGENT

**Agent architecture**   The agent is composed of a stochastic actor which samples actions from a learned policy with parameter vector $\theta$, and a deterministic critic which predicts the expected discounted sum of future rewards the actor will achieve from the current state with parameter vector $\psi$. Both the actor and critic condition only on the current model state $\hat{s}_t$, which is Markovian:

$$
\begin{aligned}
&\text{Actor:} && a_t \sim \pi_\theta(a_t \mid \hat{s}_t) \\
&\text{Critic:} && v_\psi(\hat{s}_t) \approx \mathbb{E}_{\pi_\theta, p_\phi}[\Sigma_{t=0}^H \gamma^t r_t]
\end{aligned}
\tag{12}
$$

We train the critic to regress the $\lambda$-target (Sutton & Barto, 2005)

$$
V_t^\lambda = r_t + \gamma \left( (1 - \lambda) v_\psi(\hat{s}_{t+1}) + \lambda V_{t+1}^\lambda \right), \quad V_{t+H}^\lambda = v_\psi(\hat{s}_{t+H})
\tag{13}
$$

which lets us control the temporal-difference (TD) learning horizon with the hyperparameter $\lambda$. Setting $\lambda = 0$ recovers 1-step TD learning, while $\lambda = 1$ recovers unbiased Monte Carlo returns, and intermediate values represent an exponentially weighted sum of n-step returns. In practice we use $\lambda = 0.95$. To train the critic, we regress the $\lambda$-target directly with the objective:

$$
\min_\psi \mathbb{E}_{\pi_\theta, p_\phi} \left[ \sum_{t=1}^{H-1} \tfrac{1}{2} (v_\psi(\hat{s}_t) - sg(V_t^\lambda))^2 \right]
\tag{14}
$$

There is no loss on the last time step since the target equals the critic there. We follow Mnih et al. (2015), who suggest using a copy of the critic which updates its weights slowly, called the target network, to provide the value bootstrap targets.

The actor is trained to maximize the discounted sum of rewards predicted by the critic. We train the actor to maximize the same $\lambda$-target as the critic, and add an entropy regularization term to encourage exploration and prevent policy collapse. We optimize the actor using REINFORCE gradients (Williams, 2004) and subtract the critic value predictions from the $\lambda$-targets for variance reduction. The full actor loss function is:

$$\mathcal{L}(\theta) = \mathbb{E}_{\pi_\theta, p_\phi} \left[ \sum_{t=1}^{H-1} \underbrace{-\log \pi_\theta(a_t \mid \hat{s}_t) sg(V_t^\lambda - v_\psi(\hat{s}_t))}_{\text{reinforce}} - \underbrace{\eta H(\pi_\theta(\hat{s}_t))}_{\text{entropy regularizer}} \right] \quad (15)$$

**Algorithm** Learning proceeds in two phases: First, we train the WM on all available demonstration data using the ELBO objective 10. Next, we encode expert demonstrations into the world model latent space, and use the on-policy actor critic algorithm described above to optimize the intrinsic reward 8, which measures the divergence between agent and expert over time in latent space. In principle, any on-policy RL algorithm could be used in place of actor-critic. We describe the full procedure in Algorithm 1.

---

**Algorithm 1** Dream Imitation (DITTO)

---

1: **Require** demonstration data $\mathcal{D} = \left\{ (x_t, a_t, x_{t+1})_{t=0}^{\|e_n\|} \mid n \in N \right\}$
2: Initialize world model parameters $\phi$
3: **while** *not converged* **do**                                       ▷ World model learning
4:     Draw $B_{wm}$ transition sequences $\{(x_t, a_t, x_{t+1})_{t=k}^{k+L}\} \sim \mathcal{D}$
5:     Compute all sequential RSSM components according to eqn 9
6:     Update $\phi$ with ELBO loss 10
7: **end while**
8: Initialize actor and critic parameters $\theta, \psi$
9: **while** *not converged* **do**                                             ▷ Agent training
10:     Draw $B_{ac}$ expert latent state sequences $(\hat{s}_\tau^E) \sim \hat{\mathcal{D}}^E$
11:     Generate trajectories $(\hat{s}_\tau^\pi, a_\tau)_{\tau=t}^{t+H}$ with $a_\tau \sim \pi_\theta(\cdot \mid \hat{s}_\tau)$
12:     Compute rewards $r_\tau^{\text{int}}(\hat{s}_\tau^\pi, \hat{s}_\tau^E)$ and values $v_\psi(\hat{s}_\tau^\pi)$
13:     Compute $\lambda$-returns $V_\tau^\lambda = r_t + \gamma \left( (1-\lambda)v(\hat{s}_{\tau+1}^\pi) + \lambda V_{\tau+1}^\lambda \right), \quad V_{\tau+H}^\lambda = v(\hat{s}_{\tau+H}^\pi)$
14:     Update critic on $\lambda$-targets: $\sum_{\tau=t}^{t+H} \frac{1}{2}(v_\psi(\hat{s}_\tau^\pi) - sg(V_\tau^\lambda))^2$
15:     Update actor with eqn 15
16: **end while**

---

## 4 EXPERIMENTS

To the best of our knowledge, we are the first to study completely offline imitation learning without behavior cloning in pixel-based observation environments. Prior works generally focus on improving behavior cloning (Sasaki & Yamashina, 2021), or study a mixed setting with some online interactions allowed (Rafailov et al., 2021) (Kidambi et al., 2021). To demonstrate the effectiveness of world models for imitation learning, we train without any interaction with the true environment.

### 4.1 AGENTS

To test the performance of our algorithm, we compare DITTO to a standard baseline method, behavior cloning, and to two methods which we introduce in the world model setting.

**Behavior cloning** We train a BC model end-to-end from pixels, using a convolutional neural network architecture. Compared to prior works which study behavior cloning from pixels in Atari games (Hester et al., 2017)(Zhang et al., 2020)(Kanervisto et al., 2020), our implementation achieves stronger results, even in games where it is trained with lower-scoring data.

**Dream agents** We adapt GAIL (Ho & Ermon, 2016) and BC to the world model setting, which we dub D-GAIL and D-BC respectively. D-GAIL and D-BC both receive world model latent states instead of pixel observations. The D-BC agent is trained with maximum-likelihood estimation on

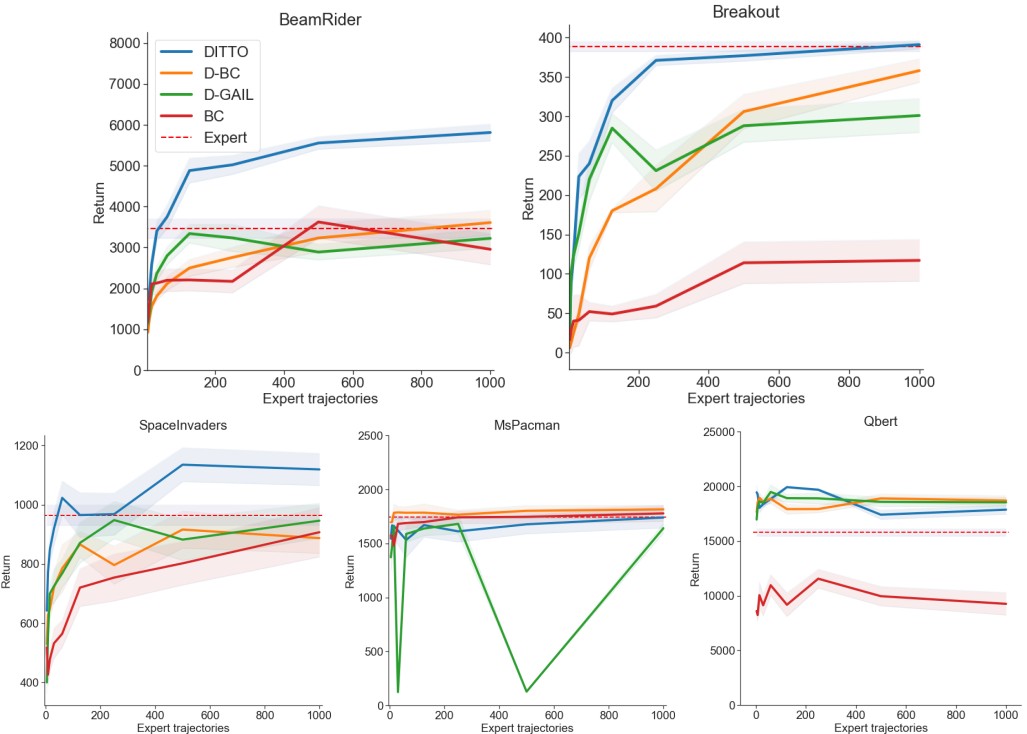

Figure 2: Results on five Atari environments from pixels, with fixed horizon $H = 15$. The plots show mean final performance across 5 runs with 20 validation simulations, and shaded regions show $\pm 1$ standard error. The experts are strong pre-trained PPO agents from the RL Baselines3 Zoo.

the expert demonstrations in latent space, with an additional entropy regularization term which we found stabilized learning:

$$L_{BC} = \mathbb{E}_{(\hat{s},a) \sim \hat{\mathcal{D}}^E} \left[ - \log \left( \pi(a|\hat{s}) \right) - \eta_{BC} H(\pi(\hat{s})) \right] \qquad (16)$$

The D-GAIL agent is trained on-policy in the world model using the adversarial objective from Equation 3. The D-GAIL agent optimizes its learned adversarial reward with the same actor-critic formulation used by DITTO, described in Section 3.3. We train both DITTO and D-GAIL with a fixed horizon of $H = 15$. At test-time, the model-based agent policies are composed with the world model encoder and RSSM to convert high-dimensional observations into latent representations.

All model-based policies in our experiments use an identical multi-layer perceptron (MLP) architecture for fair comparison in terms of the policies' representation capacity, while the BC agent is parameterized by a stacked CNN and MLP architecture which mirrors the world model encoder plus agent policy. We found that D-GAIL was far more stable than expected, since prior works (Reddy et al., 2020) (Brantley et al., 2020) reported negative results training GAIL on Atari games from pixels in the easier online setting. This suggests that world models may be beneficial for representation learning even in the online case, and that other online algorithms could be improved by moving them to the world model setting.

We evaluate our algorithm and baselines on 5 Atari environments, using strong PPO agents (Schulman et al., 2017) from the RL Baselines3 Zoo (Raffin, 2020) as expert demonstrators, using $N^E = \{4, 8, 15, 30, 60, 125, 250, 500, 1000\}$ expert episodes to train the agent policies in the world model. To train the world models, we generate 1000 episodes from a pre-trained policy, either PPO or advantage actor-critic (A2C) (Mnih et al., 2016), which achieves substantially lower reward compared to PPO. Surprisingly, we found that the A2C and PPO-trained world models performed similarly, and that only the quality of the imitation episodes affected final performance. We hypothesize that this is because the A2C and PPO-generated datasets provide similar coverage of the environment. It appears that the world model can learn environment dynamics from broad classes

of datasets as long as they cover the state distribution well. The data-generating policy's quality is relevant for imitation learning, but appears not to be for dynamics learning, apart from coverage.

## 4.2 RESULTS

Figure 2 plots the performance of DITTO against our proposed world model baselines and standard BC. In MsPacman and Qbert, most methods recover expert performance with the least amount of data we tested, and are tightly clustered, suggesting these environments are easier to learn good policies from low amounts of data. D-GAIL exhibited adversarial collapse twice in MsPacman, an improvement over standard GAIL, which exhibits adversarial collapse much more frequently in prior works which study imitation learning from pixels in Atari (Reddy et al., 2020)(Brantley et al., 2020). In contrast, DITTO always recovers or exceeds average expert performance in all tested environments, and matches or outperforms the baselines in terms of both sample efficiency and asymptotic performance.

## 5 CONCLUSION

Addressing covariate shift in imitation learning is a long-standing problem. In this work we proposed DITTO, a method which addresses this problem in the challenging offline setting where no environment interactions are allowed while learning. The offline setting exacerbates covariate shift, since the agent learns from and acts under unrelated distributions, and cannot easily estimate its own induced state distribution to perform off-policy corrections. Recent strong IL methods such as DRIL (Brantley et al., 2020) and V-MAIL (Rafailov et al., 2021) achieve excellent sample efficiency in terms of expert demonstrations, but still require environment interaction to perform on-policy learning. Our agent consistently recovers expert performance without online interaction in the tested Atari environments from pixel observations, and matches or exceeds the performance of strong baselines which we implement in the world model setting.

DITTO learns by decomposing imitation learning into two parts: First, a world model is learned to approximate the underlying environment dynamics from all available demonstration data, regardless of the quality of the policy which generated it. Next, expert demonstrations are encoded into trajectories in the world model latent space. Finally, DITTO produces on-policy rollouts in the world model, and optimizes an intrinsic reward which measures the agent's drift from expert trajectories. By optimizing this reward with standard on-policy reinforcement learning algorithms, DITTO learns to recover from its own mistakes across multiple time-steps, and match the expert state distribution, which we prove bounds the agent-expert return difference in the true environment.

Decoupling dynamics and policy learning lets us learn the world model from any historical demonstration data, and provides us with a surprisingly rich signal for imitation learning: the latent representations learned via generative world modeling. Our latent space reward extends the recent work on imitation by reinforcement learning (Ciosek, 2022) to difficult pixel-based observation environments, and contrasts with recent IL methods which need to employ adversarial (Fu et al., 2018) or sparse rewards Wang et al. (2019) to induce imitation, which can be difficult to train. Furthermore, other methods from reinforcement and imitation learning are orthogonal to and compatible with ours, and could be combined with DITTO to achieve greater performance and sample efficiency, e.g. by adding behavioral cloning rewards or uncertainty costs to DITTO's objective.

## 6 REPRODUCIBILITY STATEMENT

In order to ensure reproducibility of the results, all hyperparameters used in our experiments are included in Appendix A. The code for the experiments will be made available for the reviewing process and we intend to release the code for the camera-ready version of the paper.

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

## A    PROOF OF DIVERGENCE REWARD BOUND

We prove a corollary of proposition 1 from Ciosek (2022). Ciosek (2022) uses many intermediate results and definitions, so we encourage the reader to reference their work while reading to understand this proof.

**Corollary A.1.** *Suppose we also have another imitation learner, which uses the same data-set of size N, and still satisfies Assumption 3, but instead trains on some other intrinsic reward, $R'_{int}$ which satisfies (for some $\epsilon > 0$):*

$$R'_{int}(s, a) = 1, \forall (s, a) \in D$$
$$0 \leq R'_{int}(s, a) \leq 1 - \epsilon, otherwise$$

*Let $\rho^J$ be the limiting state-action distribution of this imitation learner. Then:*

$$||\rho^J - \rho^E||_{TV} \leq \frac{\eta}{\epsilon}$$

$$\mathbb{E}_{\rho^J}[R] \geq \mathbb{E}_{\rho^E}[R] - \frac{\eta}{\epsilon}$$

*Proof.* Lemma 5 trivially still holds with $R'_{\text{int}}$ instead of $R_{\text{int}}$, as $R'_{\text{int}} \geq R_{\text{int}}$ always, $\forall \rho, \mathbb{E}_\rho[R'_{\text{int}}] \geq \mathbb{E}_\rho[R_{\text{int}}]$. Hence the bound holding true for $\mathbb{E}_{\rho^I}[R_{\text{int}}]$ implies it holds for $\mathbb{E}_{\rho^I}[R'_{\text{int}}]$ too.

Lemma 7 holds with $\kappa$ replaced by $\frac{\kappa}{\epsilon}$, so the result is $\mathbb{E}_{\rho^I}[R] \geq (1 - \frac{\kappa}{\epsilon})\mathbb{E}_{\rho^E}[R] - 4\tau_{\text{mix}}\frac{\kappa}{\epsilon}$. We do this by considering their proof in Appendix D. The properties of the intrinsic reward are utilised in just one paragraph, after equation 25. This is done in stating that $\sum_\ell \frac{\ell M_\ell}{T} \to \mathbb{E}_{\rho^I}[R_{\text{int}}]$ and $\frac{B+1}{T} \to 1 - \mathbb{E}_{\rho^I}[R_{\text{int}}]$. This is not true for $R'_{\text{int}}$. Let $p_a$ be the limiting chance of the expert agreeing with the $R'_{\text{int}}$ imitation agent. Almost by definition, $\sum_\ell \frac{\ell M_\ell}{T} \to p_a$ and $\frac{B+1}{T} \to 1 - p_a$.

Note that $\mathbb{E}_{\rho^I}[R'_{\text{int}}] \leq p_a + (1 - p_a)(1 - \epsilon)$; we yield a reward of 1 every time we agree, and at most $1 - \epsilon$ if we disagree. Hence, using $1 - \kappa = \mathbb{E}_{\rho^I}[R'_{\text{int}}]$, we have $1 - \kappa \leq p_a + (1 - p_a)(1 - \epsilon) = 1 - \epsilon + p_a\epsilon$, hence $p_a \geq 1 - \frac{\kappa}{\epsilon}$.

So, taking limits as done in the original proof, we have:

$$\mathbb{E}_{\rho^I}[R] \geq p_a\mathbb{E}_{\rho^E}[R] - (1 - p_a)4\tau_{\text{mix}} - 0$$
$$= p_a \geq p_a(\mathbb{E}_{\rho^E}[R] + 4\tau_{\text{mix}}) - 4\tau_{\text{mix}}$$
$$\geq (1 - \frac{\kappa}{\epsilon})(\mathbb{E}_{\rho^E}[R] + 4\tau_{\text{mix}}) - 4\tau_{\text{mix}}$$

Now, combining these lemmas is exactly as in section 4.4 in Ciosek (2022). The factor of $\frac{1}{\epsilon}$ carries forward, yielding $\mathbb{E}_{\rho^J}[R] \geq \mathbb{E}_{\rho^E}[R] - \frac{\eta}{\epsilon}$ as required.

$\square$

## B  HYPERPARAMETERS

Table 1: Experimental hyperparameters

| Description | Symbol | Value |
| --- | --- | --- |
| Number of world model training episodes | $N$ | 1000 |
| Number of expert training episodes | $N^E$ | $\{4, 8, 15, 30, 60, 125, 250, 500, 1000\}$ |
| World model training batch size | $B_{wm}$ | 50 |
| World model training sequence length | $L$ | 50 |
| Agent training batch size | $B_{ac}$ | 512 |
| Agent training horizon | $H$ | 15 |
| Discount factor | $\gamma$ | 0.95 |
| $TD(\lambda)$ parameter | $\lambda$ | 0.95 |
| KL-Balancing weight | $\beta$ | 0.1 |
| KL-Balancing trade-off parameter | $\delta$ | 0.8 |
| Actor-critic entropy weight | $\eta$ | $5 \times 10^{-2}$ |
| Behavior cloning entropy weight | $\eta_{BC}$ | 0.1 |
| Optimizer | - | Adam |
| All learning rates | - | $3 \times 10^{-4}$ |
| Actor-critic target network update rate | - | 100 steps |

## C  TIME HORIZON ABLATION

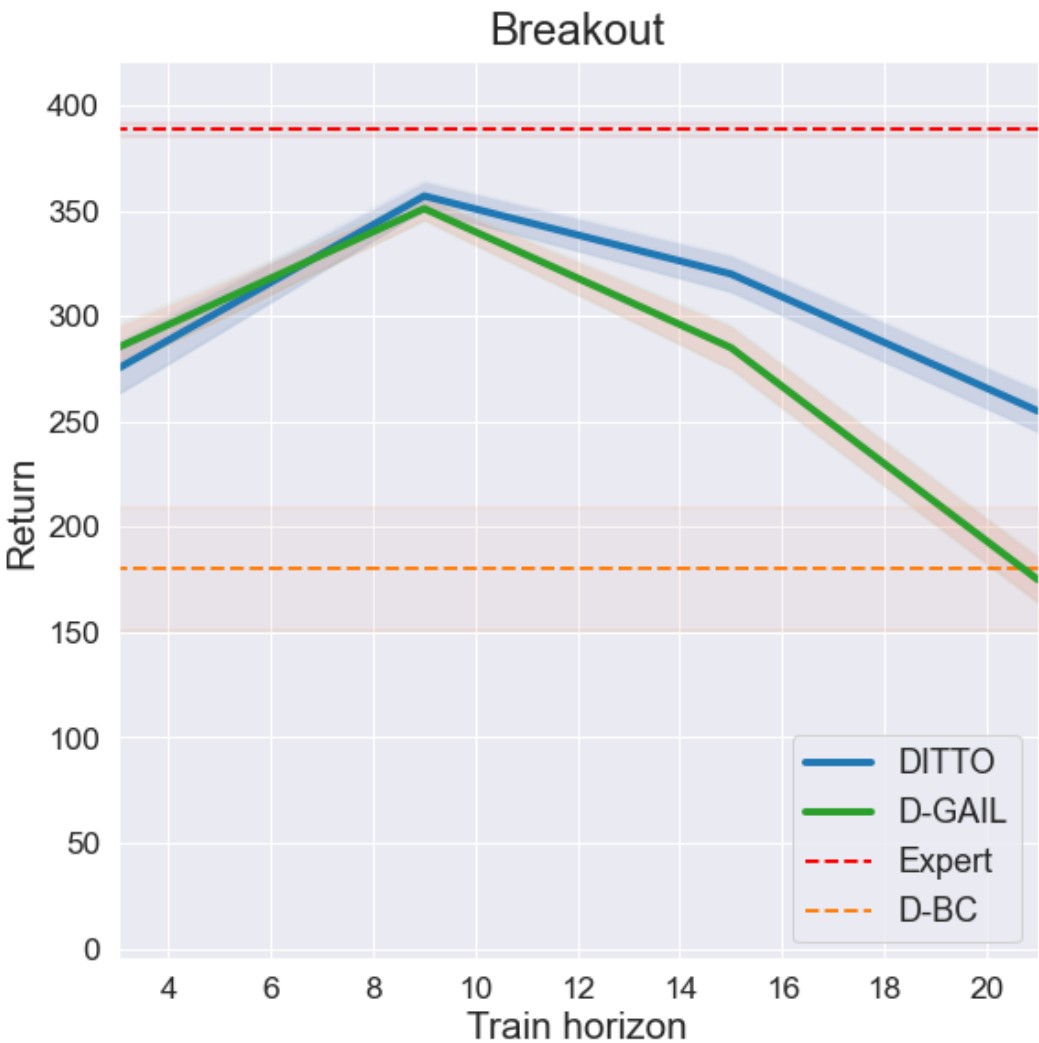

Figure 3: We ablate the agent learning horizon $H$ at fixed data, for $N = 125$ episodes. We find that the environment seems to have a characteristic planning horizon, which for Breakout appears at $H = 9$. This result suggests that learning to act over multiple steps is beneficial in some environments. D-BC is always trained to predict the next expert action, and has no planning horizon.

