# OpenReview forum: "DITTO: Offline Imitation Learning with World Models"
_ICLR.cc/2023/Conference — Submitted to ICLR 2023_

### Official Review · Reviewer_pKRJ · 2022-10-17

**Confidence:** 4
**Correctness:** 4
**Technical Novelty And Significance:** 3
**Empirical Novelty And Significance:** 3
**Recommendation:** 6

**Clarity, Quality, Novelty And Reproducibility:**

The paper is well written and motivated. The proposed method appears to be a novel combination of various existing components. The practical algorithm is described in details with source code included for reproducibility.

**Strength And Weaknesses:**

Strength:
1. The paper is well written and easy to follow. All elements of the proposed model appear well-motivated. The proposed method is sound, and appears a novel combination of various techniques.

2. Empirically, the evaluation shows the proposed method outperforming the baselines appropriately adapted to the world-model setting.

Weakness:
1. The empirical evaluation, while carefully designed, is limited in scope and does not explicitly highlight/explain why the proposed method outperforms the baselines. For instance, my understanding is that the only difference between GAIL and DITTO is that GAIL uses adversarial reward. However, from this singleton comparison it is difficult to assess if adversarial reward is worse than intrinsic reward. Related to this, there are various recent adversarial formulation that outperforms GAIL and they should also be considered in the evaluation to more broadly evaluate the impact of the reward formulation. As an example, PWIL [1] is fairly similar to DITTO in terms of how reward is computed.

2. The quality of the world-model and reward functions are not independently analyzed. In the current experiments, the size of the dataset directly affects both the quality of the world model and of the reward. However, it is important to separate these two effects, as the authors also argued that learning the world model can be done using transitions of any quality. Potential ablation may include separate the datasets used for world model and reward computation to isolate the two effects.

3. The method appears directly applicable to Mujoco environments. It would be nice to see if world model is still beneficial in a vector-based observation/state setting.

Overall, any additional experiments to improve our understanding of the proposed method and why it outperforms existing ones would be beneficial.

[1] Primal Wasserstein Imitation Learning, Dadashi et al. ICLR 2021

**Summary Of The Paper:**

The paper proposes an offline imitation learning (IL) method supported by a word model. The world model is obtained by training on the transitions collected in a dataset, without any direct access to the actual environment. A intrinsic reward is also designed that measures the divergence between expert and agent in the trained word model. The approach is well motivated. Empirical results shows that the proposed method outperforms baselines adapted to the world-model, namely GAIL and BC.

**Summary Of The Review:**

The paper proposes a fully offline IL method by training a world model and leverage standard RL for imitation learning. The method is well motivated and shows improved performance over baselines. Additional ablation and experiments are needed to better understand the advantages of the proposed method.

---

> ### Author Response · Authors · 2022-11-19
> **Response to Reviewer pKRJ**
>
> Thank you for your review.
>
> You are correct that the difference between GAIL and DITTO is that GAIL uses a learned adversarial reward, while DITTO uses a fixed intrinsic reward defined on the model latent space. Learned rewards can fail to generalize off the distribution they are trained on, and GAIL in particular is difficult to train on pixel-based environments, even in the easier online setting, as demonstrated by [1][2], who both report negative results with GAIL on pixel-based environments.
>
> One clarification, what we had labeled as “GAIL” in our plots is actually not the usual GAIL, but GAIL modified for the world model setting (or D-GAIL as we’re calling it). D-GAIL should be considered a contribution in its own right, since it actually substantially outperforms online GAIL in performance and stability, while remaining completely offline. This demonstrates the effectiveness of transforming even online problems with difficult observation spaces like Atari to the offline world model setting!
>
> Unfortunately we were not able to ablate the effect of world model training data size, as the world models take multiple days to train, so ablating this hyper parameter is too expensive for our computational budget. We have reworded our claim about the world model’s insensitivity to the quality of the data-generating policy to focus on data coverage. As mentioned in the paper, we found that training the world models from A2C vs PPO agents did not result in any meaningful performance changes, even though the PPO agents achieve far higher rewards. We expect that this is because the A2C agent data still covers the environment. The experiments in the paper show world models trained with A2C data, and data for imitation reward from PPO agents, which demonstrates that they can be separated.
>
> We agree that testing world models in vector-based environments would be interesting, but we restrict our study to the difficult “from pixels-alone” case here. We did add a behavior cloning baseline from pixels, as opposed to behavior cloning in the world model, which we mistakenly called BC before but have relabeled (D-BC) and which should be considered a contribution on its own.
>
> We have substantially expanded the theory section to discuss how our reward formulation induces imitation learning, and added a new proof of that fact. We also re-organized much of the paper to better highlight our central contribution, which is that offline imitation learning can be converted into on-policy reinforcement learning in a world model, using a simple reward on the model latent space.
>
> [1] Reddy, Siddharth et al. “SQIL: Imitation Learning via Regularized Behavioral Cloning.” ICLR 2020.
> [2] Brantley, Kianté et al. “Disagreement-Regularized Imitation Learning.” ICLR 2020.

---

### Official Review · Reviewer_ZmXh · 2022-10-25

**Confidence:** 4
**Correctness:** 4
**Technical Novelty And Significance:** 3
**Empirical Novelty And Significance:** 3
**Recommendation:** 6

**Clarity, Quality, Novelty And Reproducibility:**

Clarity

The paper is clear enough, but a bit verbose in places (eqns 11-14 could be cut, for instance)

Quality

The paper is relatively high quality, and the work is quite polished.

Novel

As far as I am aware, this approach is novel. There does seem to be a bit of similarity to the EDM approach, which might be worth clarifying.

Reproducibility

Code is provided, so the approach seems reproducible.


**Strength And Weaknesses:**

Strengths
+ The proposed method is very neat, and explained well. The improvement obtained from an explicit model-based approach is great to see, and encouraging for model-based RL in general.
+ The theory is nice to have, as is the discussion of the fundamental challenges facing imitation learning.
+ The experimental results seem promising to start with.

Weaknesses
+ The authors write 'to the best of our knowledge we are the first to study completely offline imitation learning without behavior cloning in high-dimensional observation environments'. In fact, there are several prior works which deal with this setting. They include ValueDice [1], IQ-Learn [2], EDM [3] AvRIL [4], all of which achieve decent performance. There are almost certainly more, of which you can probably identify most by looking at the papers that cite [3] .

  [1] Kostrikov et al, Imitation Learning via Off-Policy Distribution Matching, ICLR 2019

  [2] Garg et al, IQ-Learn: Inverse soft-Q Learning for Imitation, Neurips 2021

  [3] Jarrett et al, Strictly Batch Imitation Learning, Neurips 2020

  [4] Chan et al, Scalable Bayesian Inverse Reinforcement Learning, ICLR 2021

  Several of these methods are non-adversarial and so do not suffer from the (very reasonable) issues raised by the authors with regards to GAIL.

+ I think I may be misunderstanding, but is there a reason that RL is required to learn the actor in the world model? It seems that if everything is controlled and latent, we can just differentiate through the policy directly (using a Gumbel-softmax) for issues with discrete distributions.

**Summary Of The Paper:**

The paper introduces DITTO, a method for fully offline imitation learning in POMDPS high dimensional observation spaces. The idea progresses by first learning a dynamics model or 'world model'. Then, an RL agent is trained in this latent space to minimize the divergence to the learned state-action density to the expert state-action density.

**Summary Of The Review:**

The paper proposes a new method for explicitly model-based offline imitation learning. The method is exciting and shows promise. However, the experimental results are weak, not evaluating on the state-of-the-art modern offline imitation learning baselines. This is understandable since the authors were not aware of these methods. However, I don't think that a paper which lacks comparison to current benchmarks should be recommended for acceptance. I would be very happy to reconsider if the authors could provide comparisons to the methods mentioned above, and demonstrate how DITTO compares.

Update after rebuttal:

Thanks to the authors for the replies and additional baselines. Based on these, I have increased my recommendation to 6.

---

> ### Author Response · Authors · 2022-11-19
> **Response to Reviewer ZmXh**
>
> Thank you for your review, and the positive comments on our work!
>
> To your main point, we have slightly misstated the scope of “offline imitation learning without behavior cloning in high-dimensional observation environments”, when we meant to focus on offline imitation learning *from pixels alone*, which is a substantially harder problem than high-dimensional MDPs where the agent has access to the true state of the environment. To back this point up, none of the imitation learning works you cite actually address offline imitation from pixel observations alone, except in one case which we discuss later. All the works you mention  either a) do not study harder pixel-based observation environments [1] or b) include *online interactions* for the pixel-based environments [2] [3].
>
> In the case of [4], they do study offline imitation learning in a single pixel-based environment, BeamRider, which we also study. It is difficult to make a direct comparison since the expert behavior they imitate is substantially weaker than ours, which means that our agent using the least amount of data we test with substantially outperforms theirs with the most amount of data tested. Again, this comparison isn’t perfectly valid since the demonstrators are not substantially similar. However, we would like to point out that our DITTO agent ends up substantially outperforming the expert demonstrator in BeamRider. We have added new content and proof to the theory section, which demonstrates that our method is mode-seeking on the expert actions, and could explain why DITTO ends up outperforming the expert in some environments such as BeamRider. Please see the revised paper for more details!
>
> We believe the work of [4] is interesting and was not known to us before, so we thank the reviewer for pointing it out! We unfortunately did not have time to include it as a baseline during this rebuttal period, but we agree that it is relevant and would be suitable as a contemporary baseline method for comparison. We did add a standard behavior cloning baseline from pixels, rather than from the world-model latent states, which we believe will help contextualize the performance of our algorithm.
>
> Regarding your question about gumbel-softmax vs RL for policy optimization: It’s true that gumbel-softmax and straight-through estimation could be use to optimize the policy, but prior work [5] found that reinforce gradients outperformed direct back-propagation in discrete control tasks. (Interestingly, we note that the Dreamerv2 GitHub was updated this year to turn off backdrop through the dynamics even for continuous control tasks, it seems that a bug in their reinforce implementation was lowering its performance relative to directly back-propping [6]). It’s an interesting question when back-propping directly through such a heavily discretized world model like ours is appropriate, and we leave that for future work.
>
> We also re-organized much of the paper to better highlight our central contribution, which is that offline imitation learning can be converted into on-policy reinforcement learning in a world model, using a simple reward on the model latent space.
>
> [5]: Hafner et al. Mastering Atari with Discrete World Models. 2020.
>
> [6] https://github.com/danijar/dreamerv2/commit/6504715606dccb9afa78554f9572ba9b923dc0f5

---

### Official Review · Reviewer_RnpT · 2022-10-25

**Confidence:** 4
**Correctness:** 2
**Technical Novelty And Significance:** 2
**Empirical Novelty And Significance:** 2
**Recommendation:** 5

**Clarity, Quality, Novelty And Reproducibility:**

Clarity:
The method is described clearly except for several details.
In Equation (5), \hat{M} is undefined before. I guess it is a typo and it should be M?
About the reward design, it is mentioned that "unlike L2, the formulation is independent of the scale of the underlying feature space". However, it is also stated that "maximizing the max-cosine objective is equivalent to minimizing the L2 norm between two vectors". So the relation between your reward formulation and L2 distance is confusing. It will be better to explain this paragraph in the Appendix with the corresponding mathematics formulas and derivations.

Quality:
Because the proposed method uses both datasets of any quality (for world model learning) and expert demonstration data (for policy learning), I'm curious how the proposed method compares with model-based offline RL algorithms, given the same dataset with trajectories of different qualities.

The statement that "the quality of the policy used to generate world model training episodes did not affect final performance" seems overclaiming to me. For example, in Atari games Montezuma's Revenge, the environment is composed of 24 rooms in the first levels. When the quality of datasets for world model learning is moderate, many rooms will not be covered in this dataset. Thus, the world model never sees some rooms during training. With great expert demonstrations covering all these 24 rooms, will the world model fail in the unseen rooms? How does the proposed imitation learning approach perform in these rooms?

It will be great to see more analysis of the proposed method. As for world model learning, how do the number and quality of datasets affect the final performance? As for policy learning, how does the agent training horizon H affect the performance? It seems surprising that the policy can unroll in the world model for H=15 steps to predict future value.

Regarding the reward design for intrinsic reward, could you compare it with some existing reward designs based on latent space distance, such as https://proceedings.neurips.cc/paper/2018/file/35309226eb45ec366ca86a4329a2b7c3-Paper.pdf? The reward in this work is to calculate the step-by-step distance, which might be too restrictive to encourage the agent exactly follow the demo. When the environment is stochasticity (not sure whether the environment is deterministic or stochastic in the current experiment), will this kind of reward design hurt the final performance?

Novelty:
The novelty is okay. Although learning the world model and reward design according to distance in latent space is not novel ideas, it is novel to combine them in the model-based imitation learning setting.

Reproducibility:
Code for experiments is provided in Appendix. So the reproducibility is okay. The main issue is to apply the proposed method further for more environments because there is no ablative study about hyper-parameters and we have no idea how to tune the method in new environments.

**Strength And Weaknesses:**

Strengths:
The method is well-motivated to study the covariate shift issue in imitation learning, which is an important and interesting problem.
The proposed method is technically solid. The learning of the world model and on-policy actor-critic is well executed to make the relatively complicated pipeline work in Atari games.
The paper writing is mostly clear and easy to follow.

Weaknesses:
The performance improvement in only three out of five Atari games is not very impressive.
The statement "address the problem of covariate shift" in the abstract is not well-supported by the experiments. Regarding the experiments, it is unclear whether the bottleneck for baselines is covariate shift, and how much covariate shift can be addressed by the proposed method.
The evaluation and analysis can be improved. For example, how does the quality of the dataset affect the quality of the world model, and how does the quality of the world model affect the final performance of imitation learning? I doubt "the quality of the policy used to generate world model training episodes did not affect final performance". See below for more detailed questions and comments.


**Summary Of The Paper:**

This paper studies imitation learning in partially observable environments. The proposed method is to learn a world model using a dataset of any quality, unroll the agent's policy in the latent space of the world model to simulate the agent's trajectory, and train the policy with a reward function to make the agent's trajectory close to the demonstration trajectory in the latent space.

Specifically, the world model follows the recurrent neural network to predict future latent states, and the world model is trained with ELBO objective, consisting of the KL divergence between the prior and posterior distributions of latent states, and reconstruction error for the raw observation. The agent is trained via the on-policy actor-critic algorithm, where the policy is executed in the latent space of the world model. The dense reward signal penalizes the distance on the latent state densities between the agent's trajectory and the expert demonstration trajectory.

Experiments are conducted on five Atari games and the proposed is compared with behavior cloning, and generative adversarial imitation learning, showing advantages over the baselines in three out of five environments.

**Summary Of The Review:**

Overall, this paper proposed a technically solid method for imitation learning, but the evaluation can be better if conducted in more environments, with more datasets, and more design choices for the reward function.

---

> ### Author Response · Authors · 2022-11-19
> **Response to Reviewer RnpT**
>
> Thank you for your review.
>
> We have substantially re-worked the paper to focus on the tension between offline learning, which results in covariate shift due to learning under a distribution other than the agent’s, and on-policy learning, which mitigates this issue but is usually coupled to online learning. World models resolve this tension by enabling on-policy learning in model, which is “offline”.
>
> We have reworded our claim about the world model quality’s independence of the data-generating policies quality. Indeed that was poorly stated. We’ve reworded this claim to focus on the *coverage* of the environment in the data, which is of course what matters for training the world model. On this point, we experimented with training the world model from both A2C agent and PPO agent-generated data, and found that this did not affect the final imitation learning results (even though the PPO agent is stronger and receives much higher reward). We hypothesize that this is because while the PPO agent is stronger, both agent’s datasets adequately cover the environment. The final imitation step is always done from a separate set of PPO episodes, which demonstrates the world model’s generalization capability outside the policy data it was trained on. We’ve clarified this in our updated submission.
>
> \hat{M} is the MDP induced by the learned transition model.
>
> We agree that our discussion of the reward was confusing! We’ve substantially reworked this section to show how we arrived at it, including adding a new proof to the appendix which proves that our reward induces imitation learning in the model. In particular, the exact choice of distance function turns out not to matter for this part of the theory, so the choice between e.g. L2 and cosine similarity vs our exact formulation is due to its empirically superior performance in our initial testing.
>
> We also would have loved to study the effect of training data size on the world model, but it’s computationally expensive to train the world models, so this ablation was outside the scope of our computation budget for this work. We wanted first to demonstrate that world models could serve as effective approximate environments to do offline, on-policy imitation in.
>
> For the training horizons, we follow [1] in training the world model for a fixed horizon of H = 50, and the agent for a fixed horizon of H = 15, but we agree that this is an interesting hyper parameter to ablate. We ran an ablation over this parameter in Breakout, and have included it in the Appendix.
>
> We have re-organized much of the paper to focus on our central contribution, which is demonstrating the effectiveness of casting imitation learning as reinforcement learning in a learned model, and giving a simple reward which provably achieves this goal. We believe that the reinforcement learning, imitation learning, and world model communities will be interested in these results.
>
> [1]: Hafner et al. Mastering Atari with Discrete World Models. 2020.

---

### Official Review · Reviewer_Nmum · 2022-10-31

**Confidence:** 3
**Correctness:** 3
**Technical Novelty And Significance:** 2
**Empirical Novelty And Significance:** 2
**Recommendation:** 5

**Clarity, Quality, Novelty And Reproducibility:**

Using intrinsic rewards as in Equation (7) is interesting and novel.

However, theoretical justification seems necessary.


**Details Of Ethics Concerns:**

-

**Strength And Weaknesses:**

## **Strength**

DITTO shows good empirical results.

## **Weakness**
**Algorithm**

- DITTO uses an intrinsic reward function. However, there are no justification to use this reward function. At least there should be some consideration of the ideal case (e.g. in theory, DITTO is equivalent to a distribution matching between current policy and expert policy)

**Experiments**

- In Figure 3. D-BC always achieves near-expert performance when the number of expert trajectories is large enough. Compared to D-BC, DITTO shows slightly better performance. Therefore, it seems that more experiments are needed to show the superiority of DITTO.
- I am not sure if it is a good thing to outperform the expert in (offline) imitation learning. Why DITTO exceeds average expert performance?

**Related Works**

- One of the simplest ways to solve offline imitation learning with learned world model is to use HIDIL [1], which aims to solve an offline imitation learning problem given a misspecified simulator.
- In addition, existing (model-free) offline IL algorithms [2-4] can also be used for policy learning on learned world model.

[1] Jiang, Shengyi, Jingcheng Pang, and Yang Yu. "Offline imitation learning with a misspecified simulator." NeurIPS. 2020.

[2] Kim, Geon-Hyeong, et al. "DemoDICE: Offline imitation learning with supplementary imperfect demonstrations." ICLR. 2022.

[3] Xu, Haoran, et al. "Discriminator-weighted offline imitation learning from suboptimal demonstrations." ICML. 2022.

[4] Ma, Yecheng Jason, et al. "SMODICE: Versatile Offline Imitation Learning via State Occupancy Matching." ICML. 2022.

**Summary Of The Paper:**

This paper introduces an offline imitation learning algorithm. The proposed algorithm consists of two parts: (1) to train a world model using demonstration of any quality and (2) to train a policy using $\lambda$-TD where the reward function is computed as Equation (7) (i.e. intrinsic reward). Finally, the authors empirically show that the proposed method outperforms baselines in offline setting.

**Summary Of The Review:**

Solving offline imitation learning is an important, but the proposed algorithm is too empirical.
This paper uses only 5 Atari domains and there is no theoretical analysis.

---

> ### Author Response · Authors · 2022-11-19
> **Response to Reviewer Nmum**
>
> Thank you for your review.
>
> We have substantially re-worked the theory section of our paper, which adds a new proof and justification for how we arrived at the reward’s exact form. We also comment on how our formulation is mode-seeking, which could explain why it outperforms the expert in some cases. Furthermore, we show how to make our algorithm distribution matching, but prefer the mode-seeking formulation since it is computationally cheaper, and works well in practice.
>
> We also added an additional baseline from pixels alone (BC), and clarified that the previous baselines were actually contributions, reformulating the popular GAIL and BC algorithms in the world model setting.
>
> We have also re-organized much of the paper to better highlight our central contribution, which is that offline imitation learning can be converted into on-policy reinforcement learning in a world model, using a simple reward on the model latent space.

---

### Author Response · Authors · 2022-11-11
**Kicking off the discussion, a response to all reviewers about the central contribution**

Thank you to all reviewers for the detailed feedback!

With the intent to get the discussion started before providing more detailed responses to the individual reviews, we would like to comment on a common thread in the reviews which we believe requires clarification:

Our work’s central contribution is demonstrating that offline imitation learning in difficult environments (e.g. Atari from pixels alone) can be successfully converted to on-policy RL in a learned world model. Further, policy learning in the world model can even *out-perform fully online learning* (e.g. our formulation of offline GAIL in the world model significantly outperforms prior online versions). To our knowledge, this way of framing the problem is unique, and we solve difficult environments, most of which have not been attempted before in the offline setting.

While DITTO remains the primary algorithmic contribution of the paper, we recognize that referring to the methods we compare DITTO to as “baselines” is a misnomer. Indeed, we show that transferring two standard IL methods, Behavior Cloning (BC) and Generative Adversarial Imitation Learning (GAIL), to the world model works. This demonstrates that the world model setting is compatible with multiple learning methods (our RL method DITTO, generative adversarial imitation learning, GAIL, and supervised imitation learning, BC). Furthermore, we show that transferring the methods to the world model setting actually improves performance. E.g. prior works [1][2] report negative results with GAIL, and substantially weaker results with BC from pixels, whereas our versions of GAIL and BC, (D-GAIL and D-BC in the paper) are almost as strong as DITTO. We posit, therefore, that the results obtained in our current “baselines” corroborate our thesis that offline imitation learning can be successfully converted to on-policy RL in a learned world model and constitute relevant contributions in their own right.

Some reviewers also comment on the theoretical motivation for our specific reward function. We demonstrate empirically that it is an effective way to formulate imitation learning as reinforcement learning in the world model latent space. However, we will add a new proof to the submission which directly connects it with the strong theoretical results from [3] to provide further motivation, and makes an interesting connection with its inner-product nature.

In recognition that our communication of the points above in the submission requires improvement, we will be reorganizing the paper to make our main contribution more clear. We will add additional content such as the proof mentioned above as well as a BC from pixels comparison as a baseline, since this is the most common method with consistent results in the literature. We look forward to the reviewers’ thoughts on the points above during this discussion phase!


[1]: Siddharth Reddy, Anca D. Dragan, and Sergey Levine. Sqil: Imitation learning via reinforcement learning with sparse rewards. In ICLR, 2020.

[2]: Kiante Brantley, Wen Sun, and Mikael Henaff. Disagreement-regularized imitation learning. In ICLR, 2020.

[3]: Kamil Ciosek. Imitation learning by reinforcement learning. In ICLR, 2022.

---

### Decision · Program_Chairs · 2023-01-20

**Decision:**

Reject

**Justification For Why Not Higher Score:**

Needs more experimental validation, expand and rewrite to make it more rigorous and test on a wider variety of environments.

**Justification For Why Not Lower Score:**

N/A

**Metareview: Summary, Strengths And Weaknesses:**

This paper proposes combining the idea of learning a world model with imitation learning to help mitigate the covariance shift problem in imitation learning. The key idea is to reformulate the imitation learning problem as an on policy optimization problem in the latent world model space. The counterpart framed in this paper to this approach is vanilla behavior cloning and GAIL (generative adversarial imitation learning). The paper does a good job at showing that this approach has promise and can do better than vanilla imitation learning. Their method is compatible with these approaches and augment them in interesting ways as shown in the experiments.

The key experimental result is to show that the world model formulation leads to better performance on 5 atari games. The writing is reasonable but needs more rigor so others can build on it.

The biggest current limitation of this paper is experimental validation. I would encourage the authors to experiment on more domains so the generality of this approach is demonstrated and validated more carefully. Using world models for imitation learning is an extremely intereting and important topic for reinforcement learning research and this approach has a lot of promise.